# Genomewide landscape of gene–metabolome associations in *Escherichia coli*

Tobias Fuhrer[†] ID, Mattia Zampieri[†], Daniel C Sévin[†,‡], Uwe Sauer[*] ID & Nicola Zamboni ID

## Abstract

Metabolism is one of the best-understood cellular processes whose network topology of enzymatic reactions is determined by an organism's genome. The influence of genes on metabolite levels, however, remains largely unknown, particularly for the many genes encoding non-enzymatic proteins. Serendipitously, genomewide association studies explore the relationship between genetic variants and metabolite levels, but a comprehensive interaction network has remained elusive even for the simplest single-celled organisms. Here, we systematically mapped the association between > 3,800 single-gene deletions in the bacterium *Escherichia coli* and relative concentrations of > 7,000 intracellular metabolite ions. Beyond expected metabolic changes in the proximity to abolished enzyme activities, the association map reveals a largely unknown landscape of gene–metabolite interactions that are not represented in metabolic models. Therefore, the map provides a unique resource for assessing the genetic basis of metabolic changes and conversely hypothesizing metabolic consequences of genetic alterations. We illustrate this by predicting metabolism-related functions of 72 so far not annotated genes and by identifying key genes mediating the cellular response to environmental perturbations.

**Keywords** functional genomics; GWAS; interaction network; metabolism; metabolomics
**Subject Categories** Genome-Scale & Integrative Biology; Metabolism; Methods & Resources
**Mol Syst Biol. (2017) 13: 907**

## Introduction

Decades of *in vitro* biochemistry have established extensive enzyme-catalyzed networks of metabolite conversions, culminating in genome-scale reconstructions of bacterial metabolism with about 1,000 reactions (Feist *et al*, 2009; Orth *et al*, 2011). Largely unexplored is the even larger network of metabolites affecting general protein activities (Heinemann & Sauer, 2010; Link *et al*, 2013) and proteins influencing metabolism, mechanistically connected through direct or indirect relationships such as regulation processes or functional consequences, respectively. Scarce molecular knowledge of these interactions limits our ability to predict how genetic perturbations propagate throughout the multiple interlinked networks and affect the global metabolic state of a cell (Wang *et al*, 2010; Ghazalpour *et al*, 2014; Shin *et al*, 2014). Although gene–gene interactions exclusively based on growth phenotype measurements have provided valuable guidance in the form of genome-scale functional interaction maps (Costanzo *et al*, 2010; Nichols *et al*, 2011), the underlying mechanisms of such genetic interactions often remain obscure.

To resolve the complex traits by which abolished gene products can influence metabolism, we here exploit the multi-feature readout of non-targeted metabolomics to systematically map gene–metabolite associations at genome-scale in the bacterium *Escherichia coli*. To this end, we developed an experimental-computational approach to quantify the strength of gene–metabolite ion interactions from high-throughput, non-targeted metabolomics (Fuhrer *et al*, 2011) data obtained from two independent clones of each of the 3,807 mutants (Table EV1) in the *E. coli* single-gene deletion collection (Baba *et al*, 2006).

Analysis of our metabolome signatures revealed the presence of local effects caused by simple enzyme-reactant relationships but also numerous strong gene–metabolite associations that cannot be explained by metabolic proximity in classical stoichiometric models. The application potential of this comprehensive resource is demonstrated by predicting functionality of genes with unknown metabolic function and identifying genes involved in the cellular response to environmental perturbations solely on the basis of the metabolome. Hence, we believe the reported empirical associations between genes and metabolites to be a unique and powerful resource to support and inspire functional genomics studies.

## Results

### Metabolome profiling of *Escherichia coli* single knockout collection

Metabolome extracts were prepared from cultures growing exponentially in mineral salts medium containing glucose and amino acids

Institute of Molecular Systems Biology, ETH Zürich, Zürich, Switzerland
*Corresponding author. Tel: +41 44 633 36 72; E-mail: sauer@imsb.biol.ethz.ch
†These authors contributed equally to this work
‡Present address: Cellzome, GlaxoSmithKline R&D, Heidelberg, Germany

and analyzed in technical duplicates by non-targeted mass spectrometry. To enable analysis of more than 34,000 injections, we used high-throughput, flow-injection analysis on an high-resolution, time-of-flight (TOF) instrument (Fuhrer *et al*, 2011). This is a chromatography-free system that is well suited for large-scale profiling of the polar metabolome but cannot resolve metabolites with similar molecular weight and is subject to misquantification or misannotation in regions of the measured spectrum that are densely crowded with peaks or in the presence of unknown metabolites. Spectral data processing identified 3,169 and 4,365 distinct mass-to-charge ($m/z$) features in negative and positive ionization mode, respectively.

Based on the measured accurate mass, a total of 3,130 ions with distinct $m/z$ could be putatively matched to expectable ions of 1,432 of the 2,028 chemical formulas listed in the KEGG *E. coli* database (Kanehisa *et al*, 2012) (Table EV2). Since metabolites with equal molecular weight are not distinguishable, these 1,432 formulas theoretically match to 2,472 metabolites. As expected, most of the potentially detected compounds relate to abundant and polar metabolites such as intermediates of primary metabolism (Fig EV1). All data were condensed in a two-dimensional gene–metabolite ion association matrix that reports relative abundances of all detectable metabolite ions in all 3,807 analyzed deletion strains (Fig 1). Modified $z$-score normalization was applied to compare ion changes across all mutants independent of ionization mode and signal intensity. Ninety-nine percent of variability between biological replicates was estimated to be smaller than a $z$-score of 2.765 (Fig EV2A). Thus, the gene–metabolite association matrix can be directly queried for reproducibly changing ions in each mutant.

The overall rearrangements of steady-state metabolite concentrations in each mutant varied greatly across genotypes, regardless of mutant growth rate (Fig EV2B). Metabolites responded to deletion of genes from essentially all functional classes, including those with unknown function (Fig 2). The largest normalized metabolic changes in individual mutants were often located 1–2 metabolites upstream of deleted enzymes (e.g., Δ*purK* in Fig 1 and chorismate biosynthesis mutants in Fig EV3A), consistent with earlier observations (Ishii *et al*, 2007; Fendt *et al*, 2010). On the other hand, some gene deletions led to a widespread alteration of several metabolites (e.g., Δ*pstS* in Fig 1). Analogously, some metabolites responded only to a limited number of genetic ablations (e.g., enterobactin) while others varied in many mutants (e.g., xanthine) (Fig 1).

## Proximity and similarity of metabolic changes across gene deletions

Generally, metabolites are expected to change in proximity of reactions that were directly affected by gene deletions because of perturbed metabolic fluxes (Fendt *et al*, 2010). We tested this

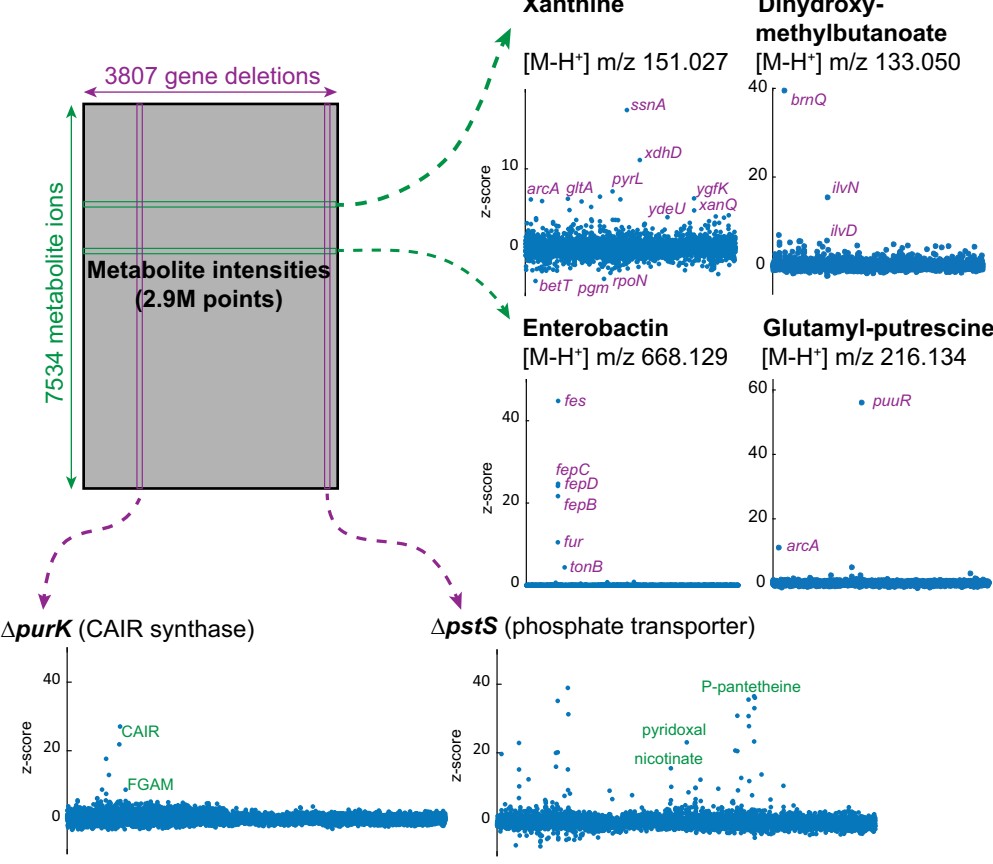

**Figure 1.  Gene–metabolite association matrix derived from metabolome analysis of single-gene deletion mutants.**

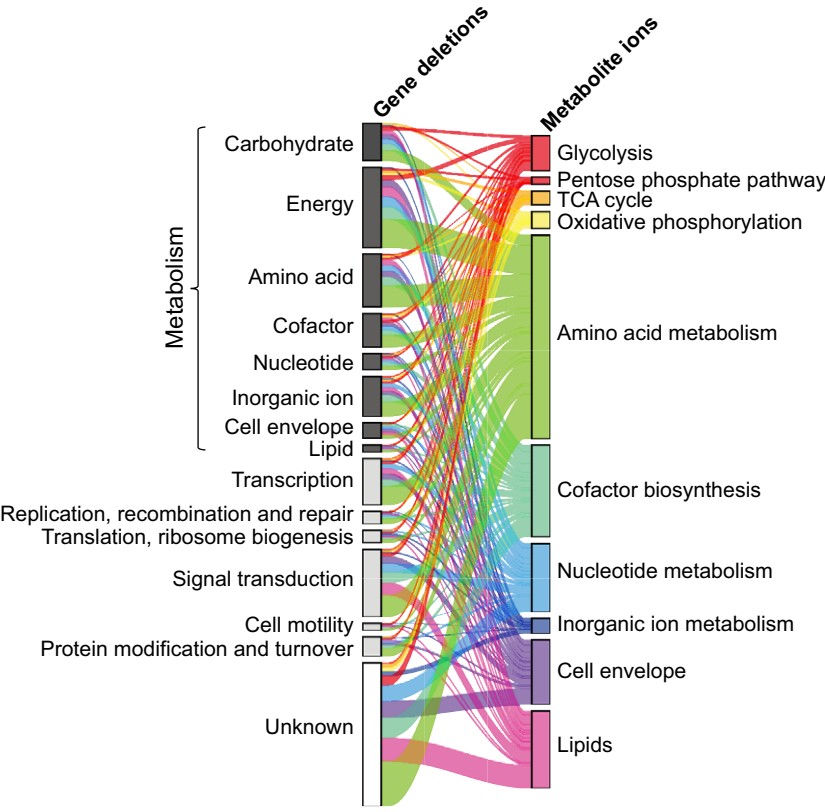

**Figure 2. Gene–metabolite associations in the 0.1% most significant associations ranked by *z*-score, corresponding to an absolute *z*-score > ~5.**
Genes were classified according to the Cluster of Orthologous Groups. Annotated ions were grouped according to the genome-scale metabolic model of *Escherichia coli* (Orth *et al*, 2011). Unknown ions were omitted. The ribbon width scales with the number of interactions.

functional association between metabolite changes and site of perturbation in mutants for which the metabolic effect can be predicted from known cellular networks. For enzyme deletions, we indeed observed a significant enrichment of metabolic changes in the immediate metabolic vicinity of the affected reactions (Figs 3, EV4 and EV5). We found a significant (*P*-value < 0.05) overrepresentation of enzyme deletions yielding the largest metabolic changes up to two enzymatic steps distance. Local metabolic changes are reabsorbed already at a distance of three, after which the reduced probability of finding the large metabolic changes remains constant.

Extending this locality analysis to larger models including the genome-scale metabolic network, the transcriptional regulatory network, and protein–protein interactions allowed us to probe the locality of measured metabolic responses of mutants lacking one of 166 transcription factors or 1,426 non-metabolic, non-enzymatic proteins, for example, those involved in regulation (e.g., PuuR) and membrane transport (e.g., BrnQ, Fig 1) (Andres Leon *et al*, 2009). In all cases, metabolite changes were enriched in the metabolic proximity of reactions that are known to be affected by the mutation (Fig EV6).

While this proximity analysis confirms the occurrence of local metabolic effects, a surprising result of this empirical genome–metabolome map is the many reproducible associations between genes and seemingly distant metabolites. Overall, genes involved in

coenzyme transport and metabolism, nucleotide transport and metabolism, signal transduction mechanisms, and transcription tend to induce widespread metabolic changes (Fig EV7). For example, we detected a strong interaction between malate and the *aro* and *pur* genes in chorismate and purine biosynthesis (Fig EV3B). Such distal changes could reflect functional interactions beyond the metabolic network topology. In most cases, the molecular links underlying such distal gene–metabolite associations remain elusive at this point, yet these interactions appear to be gene-specific rather than an unspecific consequences of mutant growth rate. One of the better understood molecular links is the strong association between the enterobactin-dependent iron uptake system (encoded by the *ent*, *fep*, and *fes* operons) and citrate/aconitate in the TCA cycle, highlighting the dichotomous role of iron as an essential modulator of aconitase activity (Varghese *et al*, 2003) and citrate as an iron chelator (Fig EV3B). These newly identified distal interactions provide leads to mechanisms of coordination across cellular pathways and functional modules.

The observed occurrence of widespread metabolic responses to specific genetic perturbations complicates the interpretation of metabolomics data, in particular from mutants lacking a gene with unknown function. We wondered whether the metabolome profiles are sufficient to characterize the genetic lesion regardless of prior information on network structure or metabolic proximity. We therefore determined the metabolome similarities between mutants

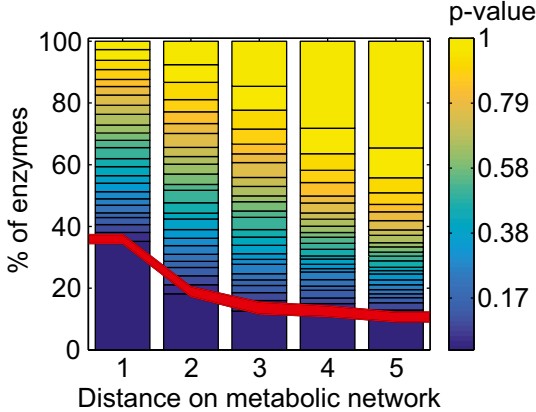

**Figure 3.  Locality analysis for enzyme deletions.**
Distribution of empirical *P*-values (calculated from a permutation test) for enzyme deletions and the respective metabolites up to a distance of five enzymatic steps are plotted. In each enzyme-deletion mutant, the modified *z*-scores of metabolites at distance 1, 2, 3, 4 or 5 are compared to the average changes generated by selecting metabolites at random. For the five tested distances between enzyme and metabolites, the fraction of enzyme deletions yielding significant distance enriched metabolic changes are highlighted below the red line. For a substantial fraction of tested enzymes, the largest metabolic changes are observed within up to two enzymatic distance steps.

lacking genes encoding for (i) complementary partners of protein complexes or (ii) isoenzymes. Although not all genes are expected to be relevant under the tested condition, we were able to recover significant fractions of both types of functional relationships using the context likelihood of relatedness (CLR) algorithm (Faith *et al*, 2007) (Fig 4A). The best-reconstructed protein complexes belonged to multi-subunit enzymes (Fig 4B), but strong similarity was also found among functionally related processes, for example, between the *sdh* operon-encoded succinate dehydrogenase and the quinol oxidase (*cyo*) or between the NADH:ubiquinone oxidoreductase (*nuo*) and the fumarate reductase (*frd*) complexes. Thus, similar metabolic profiles indeed reflect functional dependencies between genes.

Our analyses demonstrate that metabolome profiles correctly capture known functional links between related genes and cellular processes, even if underlying gene–metabolite associations go beyond the canonical metabolic network. Consequently, our empirically constructed association map provides a unique resource for data-driven investigation of gene–metabolite interactions. In the following, we provide two representative applications to illustrate how the association map can predict metabolic function of orphan genes and identify potential genes mediating metabolic adaptations to external perturbations.

### Prediction of orphan gene function

A particularly persisting problem in the post-genomic era is the 30–40% fraction of genes with unknown function (y-genes) even in the best-characterized species (Jaroszewski *et al*, 2009; Hanson *et al*, 2010). To demonstrate the use of the association map in detailing the roles of uncharacterized genes, we attempted to infer functions for the 1,274 y-genes (Hu *et al*, 2009) in our screen by searching for

similarities between the metabolome profiles of their deletion mutants and those of the 2,533 mutants lacking genes with known functions. Metabolic and other cellular functions were enriched among deleted genes eliciting similar responses for one quarter and about half of the y-gene mutants, respectively (Fig 5A). Based on consistency of enriched pathways among similar responding mutants and the observed differential ions in y-gene knockouts, we propose metabolism-related functions for 72 y-genes (Table EV3). Enrichment scores of enzymes or transcription factors among the similar genes were mutually exclusive, indicating that strong local and weaker global effects form two separate groups also among orphan genes (Fig 5B).

Two of our top y-gene predictions, *Yhhk* and *YgfY*, were annotated during the course of this study. These are representative examples to illustrate how functional hypotheses are derived from the association map and similarity analysis (Fig EV8A–C). *YhhK* knockout mutant exhibited strong similarity to several enzymes in pantothenate and coenzyme A biosynthesis (*panB*, *panC*, *panD,* and *panE*), with several differential metabolites in the coenzyme A biosynthesis pathway, including (R)-pantoate and (R)-pantothenate. Consistent with our data, YhhK was found to be an acetyl-transferase activating PanD and recently annotated as PanZ (Nozaki *et al*, 2012; Stuecker *et al*, 2012). Similarly, metabolome profiling of *ygfY* deletion mutant featured common metabolic changes (Table EV3) to genes encoding subunits of succinate dehydrogenase (i.e., *sdhB* and *sdhC*). In addition, the top ranked differential ions observed in the *ygfY* mutant are dominated by succinate annotated ions (Table EV3). While enzymatic assays of YgfY using succinate as only substrate did not result in any detectable catalytic activity, when using crude cell lysate of the *ygfY* mutant instead of purified protein, we found succinate dehydrogenase activity to be drastically reduced in comparison with wild type (Fig EV8B). In addition, growth of the *ygfY* mutant was almost completely abolished when cells were grown on succinate as sole carbon source (Fig EV8C). Consistent with our findings, *ygfY* was recently reannotated to encoded sdhE, a FAD assembly factor for SdhA and FrdA (McNeil *et al*, 2012, 2013, 2014).

Different from the two previous examples, the functional role of YidK and YidR is still unknown. Metabolome-based predictions suggested a common role of these two genes in galactose and gluconate/galacturonate metabolism, respectively (Fig EV8D and E). Deletion mutants of functionally characterized genes with similar metabolic responses are mainly related to sugar catabolism (e.g., *treF*, *malS,* and *tktA*). Furthermore, most strongly affected metabolites include various sugar derivates including UDP-galactose, 3-deoxy-D-manno-2-octulosonate, or tagatose 6-phosphate (Table EV3). These observations are consistent with other large-scale datasets such as StringDB (Szklarczyk *et al*, 2011) suggesting a genomic context-based relation to galactose metabolism, or M3D showing good expression level correlation with genes involved in carbohydrate metabolism such as *fucR*, *fucI*, *fucK*, *xylAB*, and *rfaB* (Fig EV8E). Because of this metabolic similarity and because of its membrane-localized domain (Reizer *et al*, 1994), we hypothesized that YidK might be involved in the transport of some sugar-related compounds. Consistent with our prediction, Δ*yidK* mutant exhibited a growth phenotype when grown in galactose (Fig EV8D).

For *yidR* gene deletion, the strongest metabotype similarity was observed for *dgoT* (galactonate transporter), and we consistently

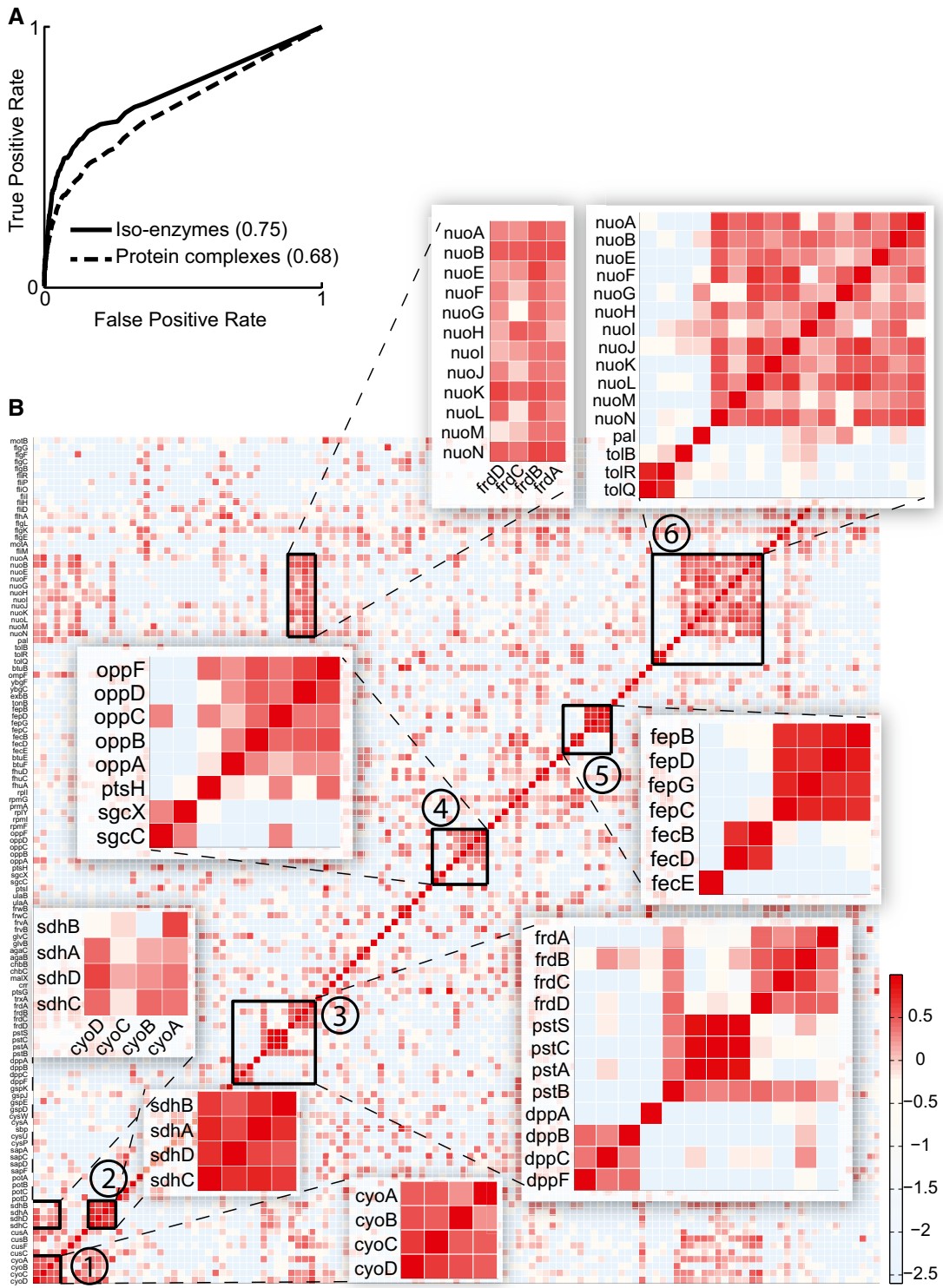

**Figure 4. Network recovery for isoenzymes and protein complexes.**

A  Recovery of enzyme function. Receiver operating characteristic curves obtained for the recovery of *Escherichia coli* isoenzymes and protein complexes based on the metabolome profiles recorded in single deletion mutants. The area under the curve (AUC) is reported in parentheses.

B  Consistent metabolic patterns in mutants of protein complex subunits. The heatmap shows the pair-wise similarity (e.g., CLR index) between metabolome response to gene deletions. Genes related to densely connected protein complexes consisting of at least three subunits are selected. We visualize the protein complex adjacency matrix, opportunely reordered. Magnified protein complexes are 1, succinate dehydrogenase; 2, cytochrome bo terminal oxidase; 3, fumarate reductase/phosphate ABC transporter/dipeptide ABC transporter; 4, murein tripeptide ABC transporter; 5, ferric enterobactin transport complex/ferric dicitrate transport system; 6, NADH: ubiquinone oxidoreductase/Tol–Pal cell envelope complex and high-scoring combinations thereof.

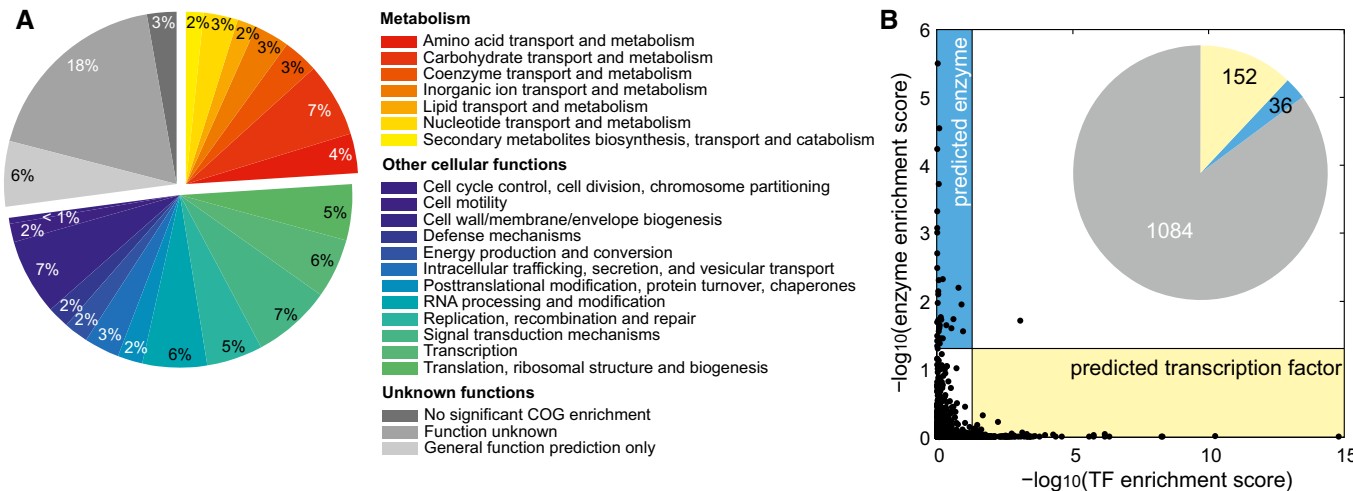

**Figure 5.  Enrichment of metabolic functions for orphan genes.**

A   Enrichment of metabolic functions (defined by Clusters of Orthologous Groups, COG) for each y-gene based on genes of known function with similar metabolome profiles, as determined by CLR.

B   Mutually exclusive function prediction of orphan genes as either enzymes or transcription factors (TF). The inset represents the number of genes predicted to be TFs (yellow), enzymes (blue), or neither (gray). One gene was predicted to be both a TF and an enzyme.

observed changes in different metabolites related to carbohydrate metabolism, such as sedoheptulose-7-phosphate or D-sorbitol 6-phosphate. Additionally, four genes of the D-galactonate degradation pathway (*dgoT*, *dgoD*, *dgoR*, and *dgoK*) were among the top ten correlating genes in the M3D database (Fig EV8E). Growth phenotyping revealed that, indeed, the *yidR* deletion mutant displays a growth defect specifically on gluconate and galacturonate (Fig EV8D), confirming that *yidR* is functionally relevant for the assimilation of these compounds. The association map thus informs on metabolism-related gene functions and can provide leads for further functional genomics investigations. In most cases, however, the prediction relates to pathways because of the complex and widespread metabolic changes we observed. More specific prediction on specific reactions or enzyme class is only possible if the immediate substrates are detectable and characterized by a particularly strong response.

**Predicting genes mediating the metabolic response to environmental perturbations**

The rapidly growing number of large-scale metabolomics studies poses a serious challenge to data interpretation, in particular when complicated patterns emerge (Sevin *et al*, 2015). Here, we investigated whether complex metabolic patterns in response to an external perturbations can be functionally interpreted with our gene–metabolite association map. To this end, we recorded the metabolome response of wild-type *E. coli* to a variety of naturally relevant nutrient limitations (e.g., phosphate, sulfur, oxygen, or iron limitation) and stresses (osmotic and deoxycholate). By comparing the metabolic response to an external perturbing agent to the metabolome profiles of individual gene deletions, we tested the ability to recover genes mediating the adaptive response of *E. coli* to sudden environmental changes (Fig EV9). In agreement with our expectations, for nutritional limitations, we consistently identified

genes directly related to the utilization of the corresponding limiting nutrient, such as iron uptake and cysteine biosynthesis in the case of iron and sulfur limitations, respectively. Deoxycholate is found in bile acids and represents a common stress factor for bacteria in the gut, such as *E. coli* (Merritt & Donaldson, 2009). However, little is known about the underlying metabolic response. Based on our genomewide metabolome map, we identified seven single-gene deletion mutants in the compendium eliciting similar metabolic changes to those observed upon exposure to deoxycholate (Fig 6A). Notably, we found that six out of these seven mutants were substantially more resistant to deoxycholate inhibition compared to the wild-type *E. coli* strain, confirming the predicted functional role of these genes in mediating the stress response to deoxycholate (Fig 6B). Five of these beneficial mutants were disrupted in the uptake of ferric enterobactin (*fepB*, *fepC*, *fepD*, and *fepG*) or the release of iron into the cytosol (*fes*). While the function of these genes might suggest a direct role of iron, growth experiments under iron limitation and deoxycholate stress demonstrated that iron depletion *per se* is not sufficient to confer deoxycholate resistance (Fig EV10A). A common metabolic feature between these mutants and deoxycholate-stressed cells was the intracellular accumulation of enterobactin (Fig 1A). While enterobactin supplementation (Fig EV10B) did not affect deoxycholate sensitivity in the wild type (Fig 6C), mutants with disrupted enterobactin biosynthesis (Δ*entF*, Δ*entB*, Δ*entC*) were surprisingly insensitive to deoxycholate stress in minimal medium containing exogenous enterobactin (Fig 6C). Hence, while the underlying mechanism remains to be elucidated, these results support the predicted functional link between enterobactin biosynthesis and deoxycholate tolerance. Overall, interrogating the metabolic response to complex perturbations using our compendium of gene deletion profiles can reveal new and non-obvious hypothesis on the molecular players that mediate the adaptive response of *E. coli* to an external stimulus.

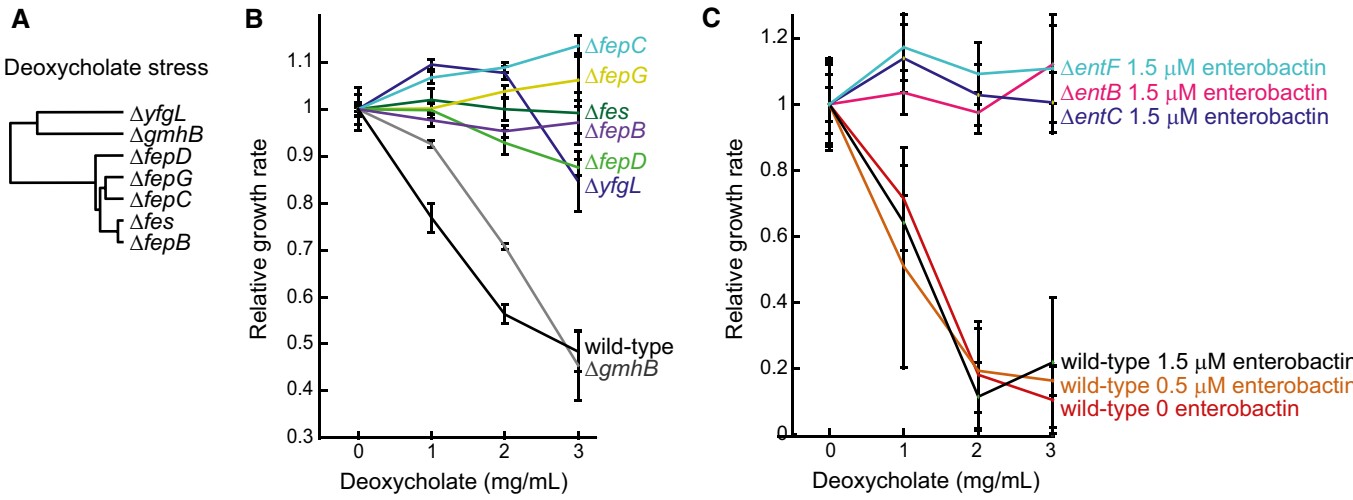

**Figure 6.  Predicting genes mediating the metabolic response to environmental perturbations.**

A  Dendrogram representing genes with significant overlap of differential metabolites in the respective knockout and during growth in the presence of 10 mg/ml deoxycholate in wild-type *Escherichia coli*. Genes are hierarchically clustered based on their topological distance assessed by the minimum number of connecting reactions in the metabolic network.

B  Relative growth rates of wild-type *E. coli* and deletion mutants in glucose minimal medium supplemented with casein hydrolysate and deoxycholate. Error bars represent standard deviations from three biological replicates.

C  Relative growth rates of wild-type *E. coli* and enterobactin biosynthesis mutants in glucose minimal medium supplemented with enterobactin and deoxycholate. Error bars represent standard deviations from three biological replicates.

# Discussion

Large-scale phenotypic screening of single- and double-gene deletion mutants has proven powerful in understanding gene functions, gene–gene interactions, and condition-dependent gene essentiality (Costanzo *et al*, 2010; Nichols *et al*, 2011). However, screening of large mutant libraries typically comes at the cost of measuring only one or few phenotypic traits (e.g., growth rate or viability). The implicit lack of fine-grained molecular or intracellular information, in turn, hinders attaining a detailed understanding on how interactions between genes, or between genes and the environment are established. Increasing the space of features that characterize functional consequences of genetic deletions on a molecular level improves the ability to disentangle the interplay between genes and to discern the regulatory architecture within cells (van Wageningen *et al*, 2010).

For the investigation of metabolism and its regulation, metabolomics provides a direct functional readout that convolutes the cell-wide interplay of enzyme activity and metabolites. Regulatory events that modify enzyme properties such as their abundance, localization, or kinetic properties eventually affect metabolite levels. Metabolomics offers the possibility of quickly profiling hundreds of metabolites involved in primary metabolism and thus characterizes the response of all key pathways that sustain growth and energy production. To this end, we generated the first comprehensive empirical map of gene–metabolite interactions by systematically measuring the relative changes in the abundance of hundreds of metabolite (ions) in 3,807 *E. coli* single-gene deletion mutants (Baba *et al*, 2006). This comprehensive compendium can be searched to find gene deletions that have the largest impact on a metabolite of interest, or vice versa, to find which metabolites are affected upon a specific gene deletion. This gene–metabolite interaction map complements classical genomewide phenotypic screens and is a valuable resource to mechanistically interpret macroscopic phenotypes.

An important result of our analysis of the gene–metabolite map is that marked metabolite changes can occur distant from the genetic lesion, even when enzymes with known catalytic functions were deleted and no global growth defect was detected. Hence, the topology and connectivity defined by the metabolic network are not sufficient to explain and predict the impact of gene deletions on the overall cell/metabolic phenotype. While these distal gene–metabolite interactions remain largely elusive to explain at this point, they can be interpreted as metabolic fingerprints of gene function and in our study were used to (i) predict the enzymatic functions of y-genes even in the absence of growth phenotypes and (ii) establish a metabolic link between gene and their functional role in mediating the cell response to environmental perturbations.

Altogether, we hypothesized metabolic function for 72 y-genes, of which YidK and YidR were experimentally validated. Our metabolome compendium also provides a key to interpret metabolic changes induced upon perturbations other than gene deletions. By establishing an indirect link between common metabolic changes induced by gene deletions and external perturbations, we predicted non-obvious mediators of the cellular response to deoxycholate treatment, which involved genes in iron metabolism.

The two above examples of functional gene annotation and predicting genes involved in cellular responses demonstrate the utility of the association map for generating novel lead hypotheses on the functional roles of genes in metabolism. Because of the large number of tested genetic lesions and covered metabolites, the map constitutes a unique resource to inspire new and less conventional

approaches to predict the mode of action of genetic and environmental perturbations. Moreover, the large number of newly revealed gene–metabolite associations paves the road to explore so far unknown functional and regulatory interactions beyond those represented in current genome-scale metabolic models (Feist *et al*, 2009; Orth *et al*, 2011).

# Materials and Methods

### Chemicals

Water, methanol and 2-propanol, all CHROMASOLV LC-MS grade, buffer additives for online mass referencing, media, and sample preparation chemicals at the highest available purity were purchased from Sigma-Aldrich and Agilent Technologies. Pure water for extraction and resuspension with an electric resistance greater than 16 MΩ was obtained from a NANOpure purification unit (Barnstead, Dubuque, IA, USA).

### Biological samples

*Escherichia coli* wild-type and 4,320 deletion mutants (Table EV1) from the KEIO knockout collection (Baba *et al*, 2006) were grown on glucose minimal medium supplemented with casein hydrolysate containing (per liter): 4 g glucose, 2 g N-Z Case Plus, 7.52 g $Na_2HPO_4 \cdot 2H_2O$, 3 g $KH_2PO_4$, 0.5 g NaCl, 2.5 g $(NH_4)_2SO_4$, 14.7 mg $CaCl_2 \cdot 2H_2O$, 246.5 mg $MgSO_4 \cdot 7H_2O$, 16.2 mg $FeCl_3 \cdot 6H_2O$, 180 μg $ZnSO_4 \cdot 7H_2O$, 120 μg $CuCl_2 \cdot 2H_2O$, 120 μg $MnSO_4 \cdot H_2O$, 180 μg $CoCl_2 \cdot 6H_2O$, 1 mg thiamine·HCl. Culture volumes of 1 ml were incubated in 96-deep well plates at 37°C with shaking at 300 rpm. Growth was followed via absorbance at 600 nm measured at four time-points, and all samples were harvested during mid-exponential growth phase by centrifugation for 10 min at 0°C and 2,200 *g*. Cell pellets were immediately extracted with 150 μl preheated water containing 2 μM reserpine and 2 μM taurocholic acid for 10 min at 80°C and occasional vortexing. This extraction broth was centrifuged for 10 min at 0°C and 2,200 *g*, and supernatants were stored at −80°C until further analysis.

### Flow-injection analysis—TOF MS

The analysis was performed on a platform consisting of an Agilent Series 1100 LC pump coupled to a Gerstel MPS2 autosampler and an Agilent 6520 Series Quadrupole TOF mass spectrometer (Agilent, Santa Clara, CA, USA) as described previously (Fuhrer *et al*, 2011). The flow rate was 150 μl/min of mobile phase consisting of isopropanol:water (60:40, v/v) buffered with 5 mM ammonium carbonate at pH 9 for negative mode and methanol:water (60:40, v/v) with 0.1% formic acid at pH 3 for positive mode. For online mass axis correction, 2-propanol (in the mobile phase) and taurocholic acid or reserpine were used for negative mode or for positive mode, respectively. Mass spectra were recorded in profile mode from *m/z* 50 to 1,000 with a frequency of 1.4 s for 2 × 0.48 min (double injection) using the highest resolving power (4 GHz HiRes). Source temperature was set to 325°C, with 5 l/min drying gas and a nebulizer pressure of 30 psig. Fragmentor, skimmer, and octopole voltages were set to 175 V, 65 V, and 750 V, respectively.

### Spectral data processing and annotation

All steps of mass spectrometry data processing and analysis were performed with MATLAB (The Mathworks, Natick, MA, USA) using functions embedded in the Bioinformatics, Statistics, Database, and Parallel Computing toolboxes as described previously (Fuhrer *et al*, 2011). Peak picking was done for each sample once on the total profile spectrum obtained by summing all single scans recorded over time, and using wavelet decomposition as provided by the Bioinformatics toolbox. In this procedure, we applied a cutoff to filter peaks of less than 500 ion counts (in the summed spectrum) to avoid detection of features that are in any case too low to deliver statistically meaningful insights. Centroid lists from samples were then merged to a single matrix by binning the accurate centroid masses within the tolerance given by the instrument resolution (about 0.002 amu at *m/z* 300). The resulting matrix lists the intensity of each mass peak in each analyzed sample. An accurate common *m/z* was recalculated with a weighted average of the values obtained from independent centroiding. Because mass axis calibration is applied online during acquisition, no *m/z* correction was applied during processing to correct for potential drifts. After merging, 3,169 and 4,365 common ions were obtained for negative and positive mode, respectively, which were annotated based on accurate mass using 3 mDa tolerance (Tables EV1 and EV2). Annotation was based on assumption that $-H^+$, $+OH^-$, and $+Cl^-$ are the possible ionization options for negative mode, and $+H^+$, $+K^+$, and $+Na^+$ for positive mode. Additional commonly observed adducts were considered as described previously in detail (Fuhrer *et al*, 2011): $^{12}C_1$-$^{13}C_1$, $^{12}C_2$-$^{13}C_2$, $-H^+ + Na^+$, $-H^+ + K^+$, $-H_2O$, $-CO_2$, $-NH_3$, $-HPO_3$, $-H_3PO_4$, $+H_2PO_4Na$, $+H_2PO_4K$, $+HPO_4Na_2$, $+HPO_4K_2$, $+(H_2PO_4Na)_2$, $+(H_2PO_4K)_2$, $+(H_2PO_4)_2NaH$, and $+(H_2PO_4)_2KH$. Putative coverage of *E. coli* metabolism is shown in Fig EV1.

### Physiology

Growth rates were calculated as the slopes of linear fits to log-transformed experimentally determined OD values. Approximate harvest OD values were estimated by the calculated growth rates and harvest time. Identification of extremely sick mutants was based on a 0.1 OD cutoff on the last two OD values as the respective calculated growth rates were not reliable. Of the 3,847 unique gene knockouts, 21 extremely sick mutants were omitted from data analysis: *aceF, carA, eptB, frmA, guaA, guaB, hflD, lpd, parC, purA, rfaI, rfaP, rfaS, rplA, ybaB, ybeY, yiQ, yjjG, yqaB, yrdA,* and *ytfE.* Furthermore, 16 mutants were removed due to injection errors during mass spectrometry analysis: *ybdG, stfQ, ybiB, mntR, ybgJ, ybaK, ybjJ, yahD, ylaB, yqcA, wcaJ, ybhR, yliJ, yaiB, ybhN,* and *ykgF.* The remaining 3,810 unique gene knockouts were further analyzed.

### Data normalization and calculation of differential ions

Preprocessing of raw mass spectrometry data consists of four steps: (i) correction of intensity drift throughout sequential injections within one plate by a low pass filter over a moving window of five injections, (ii) correction of intensity drifts across plates due to regular instrument cleaning procedures, extraction effects or long-term ionization drifts by normalizing the mean of every ion to be equal over all plates, (iii) correction of intensities for harvest OD dependencies

using a locally weighted scatterplot smoothing (LOWESS). This procedure was used to perform a model-free estimation of the ion-specific dependency with harvest ODs. The local least square regression was employed to normalize for OD effects as follows:

$$\text{intensity}_{\text{ion}\,i} = \text{raw intensity}_{\text{ion}\,i} - \text{trend}_{\text{ion}\,i} + \text{median intensity}_{\text{ion}\,i}$$

(iv) a modified *z*-score is then used to select for significantly and specifically affected ions according to:

$$z\text{-score}_{(i,j)} = \frac{\text{intensity}_{(i,j)} - \text{median}_{(i,\text{all})}}{\text{std}_{(i,\text{all})}}$$

where *i* and *j* denote ion and samples, respectively, and median as well as standard deviation (std) refers to all intensities of ion *i* in the entire dataset. Modified *z*-scores referring to technical and biological replicates are summarized into a unique median modified *z*-score, and three additional mutants were removed from the dataset having inconsistent *z*-score among the replicates (*flhC*, *hrpB*, and *yhfW*), resulting in a final data set with 3,807 mutants. Notably, no internal standards were used because they are not suited to normalize thousands of mostly unknown chemical entities.

### Reproducibility between biological replicates

For each knockout mutant, two different clones contained on separate plates in the library were separately processed on different days. We assessed reproducibility of the metabolome profiles between the two biological replicates by calculating the absolute difference between *z*-scores (Fig EV2A). The resulting distribution of *z*-score differences shows that above a *z*-score of 2.765, we have 1% probability of having false positives.

### Network recovery from pair-wise similarity

Pair-wise similarity among normalized metabolome profiles for each tested mutant was calculated by the CLR approach (Faith *et al*, 2007), which estimates a similarity score for each pair of gene profiles by comparing a joint likelihood measure based on mutual information. Isoenzymes were identified by the Orth genome-scale model (Orth *et al*, 2011). Protein complexes were obtained from EciD (Andres Leon *et al*, 2009). We then evaluated the overlap between the above known interaction graphs and the inferred network of similarities derived from metabolome profiles using receiver operating characteristic (ROC) curves. Briefly, this framework allows us to span the entire range of cutoff values for the estimated pair-wise similarities to describe the trade-off between sensitivity and the false-positive rate (FPR) in the network reconstruction. Sensitivity is defined as the fraction of the known interactions, which were also inferred by our similarity score (true positive, TP) and the total number of known interactions, given by the sum of TP and true negatives (TN) [sensitivity = TP/(TP + FN)], while FPR is the fraction of incorrect inferred relationships (false positive, FP) over the sum of all known negative interactions (given by the sum of FP and true negative, TN) [FPR = FP/(FP + TN)]. Finally, the area under the curve (AUC) was used to give an estimate of the quality of the reconstruction. For Fig 4A, the

analysis was performed by restricting the calculation of TP and FP interactions only among 46% of genes with at least one significant metabolic change (i.e., silent genes were excluded, Fig EV7C).

### Locality analysis

A genome-scale network model of *E. coli* metabolism was used to determine the distance between each enzyme–metabolite pair. The resulting pair-wise distance matrix between metabolic enzymes and metabolites was estimated by means of the minimum number of reactions separating the two in a non-directional network. All highly connected metabolites were removed prior to calculation. To assess whether largest metabolic changes are statistically more probable in the proximity of the deleted enzymes (Fig 3), we used a permutation test. For each enzyme deletion, we calculated the sum of absolute changes of metabolites directly linked to the enzyme (i.e., substrates/products) corresponding to distance 1, up to metabolites at five enzymatic steps away from deleted gene. For each tested enzyme, the observed statistic ($S_{\text{obs}}$) was compared with a permuted one ($S_{\text{perm}}$) obtained by randomizing 1,000 times the original distance matrix. *P*-value was empirically estimated as follows:

$$P\text{-value}_{g,d} = \frac{(S_{\text{perm}} \geq S_{\text{obs}})}{1,000}$$

where *g* is the selected gene and *d* is the distance (from 1 to 5).

This first statistical analysis revealed a tendency of enzyme deletions, often within amino acids biosynthetic pathways, to exhibit larger metabolic changes within one to two enzymatic steps (Figs 3 and EV4). This result enabled us to generalize and extend our analysis to non-metabolic genes using a locality scoring function as follows.

For transcription factors (TFs), we augmented the metabolic network with the connections between TFs and their known metabolic enzyme targets extracted from RegulonDB (Salgado *et al*, 2013). Similarly, to calculate the distance between non-metabolic proteins and detectable compounds, we included known protein–enzyme interactions from the EciD database (Andres Leon *et al*, 2009) (Fig EV6). The distance for all pairs of genes and detectable compounds was calculated as the minimum number of edges (regulatory links) connecting the non-metabolic proteins to enzymes, and enzymes to metabolites (reactions). We then implemented a locality scoring function weighted for distance as follows:

$$S(g_i) = \frac{\sum_{m=1}^{M} D_{i,m}^{-2} \cdot \left(1 - z.pval_{i,m}\right) \cdot |Z_{i,m}|}{\sum_{m=1}^{M} D_{i,m}^{-2} \cdot \left(1 - z.pval_{i,m}\right)}$$

$$S_{\text{rand}}^{k}(g_i) = \frac{\sum_{m=1}^{M} D_{\text{rand}_{i,m}^{-2}}^{k} \cdot \left(1 - z.pval_{i,m}\right) \cdot |Z_{i,m}|}{\sum_{m=1}^{M} D_{\text{rand}_{i,m}^{-2}}^{k} \cdot \left(1 - z.pval_{i,m}\right)}$$

$$P.\text{value}(g_i) = \frac{\sum_{k} \left(S_{\text{rand}}^{k} \geq S\right)}{K}$$

where for each gene (encoding for enzymes, TFs, or proteins) $g_i$, a weighted mean over corresponding $Z_{i,m}$ (Z-scores of metabolome

profiles corresponding to gene $i$ and metabolite $m$) is computed. Weights are a function of the inverse of the squared distance between the gene $i$ and the metabolite $m$ and the $z$-test $P$-value associated with the ion measurements (measure of confidence of metabolite measurements). We performed a permutation test with randomly shuffling of the distance matrix $D$ ($D_{rand}$) $K$ times (i.e., $K = 100$) to assign a significance value to each gene.

### Pathway enrichment analysis

Enzyme deletions and annotated metabolites were grouped according to the corresponding metabolic pathways as defined in the genome-scale model of Orth *et al* (2011). Only the largest 0.1% of metabolic changes were considered. The statistical significance of enzymes belonging to a metabolic pathway to elicit metabolic changes in each different metabolic pathway (Fig EV5) was assessed by a permutation test, where the metabolite and gene orders are randomly shuffled and the number of randomly assigned metabolite with a relative change in absolute abundance (modified $z$-score) > 5 is counted. Significance of the observed statistics was assessed by counting how many times randomly assigned metabolites exceed the originally observed metabolic changes in each metabolic pathway.

### Categorization of mutants based on number of differential ions and enrichment of cellular functions

We applied stringent absolute 0.1 percentile (0.1%) cutoff to the distribution of the $z$-scores to select significantly differential ions. Choosing a typical $P$-value cutoff of 0.05 for 0.1 percentile cutoff leads to four distinct scenarios: 0 hits for silent mutations, a region where we get less hits than expected (1–4 hits), a region with significant probability to get hits (5–10 hits), and a region where we get more hits than expected (> 10 hits) (Fig EV7B). Based on these distributions, we classified the strains as mutants being silent, or having rare, moderate, and global effects, respectively (Fig EV7C). Enrichment significance ($P$-value) for of Cluster of Orthologous Groups (Tatusov *et al*, 2003; Hu *et al*, 2009) within the different categories silent, rare, moderate, and global was derived by hypergeometric probability density function for each cellular function category for the 0.1 percentile cutoff (Fig EV7D).

### Potential function predictions for orphan genes—calculation of prediction score

Metabolome profile similarity based on CLR enabled us to relate genes with similar catalytic activity (Fig 4A). Hence, we envisaged the possibility to employ such similarity patterns to infer potential catalytic properties of functionally uncharacterized proteins. Moreover, the observed locality of metabolic responses can suggest potential reactants of eventually catalyzed reactions. In order to predict the enzymatic activity of a functionally uncharacterized protein, we thus combined the information of both CLR and differential annotated ions. To allow for contribution of more subtly affected differential ions, the percentile cutoff on the $z$-score was relaxed to 0.5% (corresponding to 0.01 $P$-value based on the probability distribution estimated in Fig EV2A). First, for a particular candidate gene, we selected among the top similar gene based on CLR (Table EV3—CLR hits, also see legend for prediction table

supplied as Word file) the annotated enzymes and extrapolated from overrepresented linked metabolites (i.e., their substrates and products enriched with $P$-value < 0.01, Table EV3—MET prediction hits, top hit and top score) when at least two enzymes were significantly similar. The genes with the highest similarity scores (i.e., highest CLR index) were selected using a threshold corresponding to a 5% FPR in the recovery of iso-enzyme network (Fig 4A). Columns CLR—top hit and top score in Table EV3 report the most similar gene by CLR and the corresponding CLR index value, respectively. Second, we tested for each annotated metabolite how likely it is that perturbations yielding most significant changes of the annotated ions were associated with deletion of those enzymes directly capable of converting the selected metabolite. For this test, we used a ROC curve analysis and ranked deletion mutants based on the $z$-score matrix. From this analysis, we derived for each ion-annotation an estimate of the AUC index representing annotation confidence (Table EV4). The changes in ion abundance (i.e., $z$-score) were then weighted by corresponding AUC indices (Table EV3, DIFF IONS hits, top hit, and top score). Third, we identified overrepresented metabolites linked to annotated enzymes either as substrates or products of catalyzed reactions and determined their intersection with metabolites identified as annotated differential ions (based on 0.5% percentile cutoff, Table EV3, MET Prediction overlap). Finally, the sum of AUC weighted $z$-scores of overlapping metabolites (Table EV3, CLR—prediction score) was used to rank all y-genes according to consistency between CLR-based predictions and observed differential annotated ions to prioritize further validations. Table EV3 also includes enrichment of KEGG pathways using hypergeometric probability tests of annotated similar genes based on differential ions, similar genes, or both (KEGG PATHWAYS by CLR, hits, top hits and top score). We also provide hyperlinks to three databases (i.e., Ecocyc, KEGG, and PortEco, Table EV3—links), and of enriched COG terms among similar genes (Table EV3—COG hits, top hit, and top score).

### Overexpression and purification of proteins encoded by y-genes

A genomewide *E. coli* gene overexpression library (Kitagawa *et al*, 2005) served as resource for expression and purification of proteins for functional validation. This library consists of 4,267 clones of *E. coli* K-12 AG1 each containing a plasmid encoding one ORF as a N-terminal His$_6$-fusion under control of the isopropyl-β-D-thiogalactopyranoside (IPTG)-inducible lac-promoter as well as chloramphenicol-acetyl-transferase providing chloramphenicol resistance as dominant selection marker. The selected expression strains were grown in 500 ml shake flasks (50 ml culture volume) on an orbital shaker for 16 h at 37°C and 300 rpm. Overexpression was performed in LB medium (10 g/l yeast extract, 10 g/l tryptone, 5 g/l NaCl at pH 7.5) supplemented with 5 g/l glucose, 20 μg/ml chloramphenicol, and 100 μg/ml IPTG. Cells were harvested by centrifugation at 2,200 $g$ and 4°C for 10 min. The supernatant was discarded, and the pellet was washed once with 500 μl (2 ml cultures) or 10 ml (50 ml cultures) 0.9% NaCl + 1 mM MgCl$_2$. For the 50 ml cultures, cells were resuspended in 4 ml of 100 mM Tris-buffer at pH 7.5 supplemented with 5 mM MgCl$_2$, 20 mM imidazole, 2 mM DTT, and 4 mM PMSF and lysed by three passages through a French pressure cell (Thermo Fisher) at 1,000 psig. Proteins were purified by immobilized metal-ion affinity chromatography using a sepharose resin with

nitriotriacetic acid groups chelating $Ni^{2+}$ ions. Imidazole and salts from the elution buffer were removed by ultrafiltration using 10 kDa MWCO centrifugal filters in 4 ml tube (50 ml cultures) or 96-well plate format (2 ml cultures) (Millipore). Proteins were resuspended in 2 mM Tris pH 7.4 and 1 mM $MgCl_2$ and stored at 4°C until further usage. Expression and purity was checked by standard Bradford assays and SDS–PAGE analysis.

**Predicting gene mediators of environmental perturbations**

For deoxycholate, the $IC_{50}$ (concentration inhibiting growth by 50%) of 10 mg/l was determined by automated microtiter-scale cultivations with a gradient of 0–20 mg/l deoxycholate. Limiting concentrations of 35 μM sulfate ($MgSO_4$), 100 nM ferric iron ($FeCl_3$), and 120 μM phosphate ($KH_2PO_4$) was determined analogously. Perturbation experiments were performed in 5 ml cultures in glucose-M9 minimal medium without casein hydrolysate at 37°C and 300 rpm. Anaerobic cultures were performed at 37°C under $N_2$ atmosphere in $N_2$ sparged septum flasks. 1 ml of cells at OD 0.5 was harvested in triplicates by fast filtration and extracted in 4 ml 40:40:20 acetonitrile:MeOH: dd$H_2O$ (vol %) for 2 h at −20°C. Extracts were dried under vacuum and resuspended in 150 μl dd$H_2O$. Metabolomics measurements were performed by flow-injection TOF-MS as described previously (Fuhrer *et al*, 2011), and for osmotic stress (500 mM NaCl), previously published data were used (Sevin & Sauer, 2014).

Metabolites undergoing significant changes upon an environmental perturbations compared to mock treated cells were defined as passing an absolute fold-change cutoff of 2 at a false-discovery rate of 0.01 relative to unperturbed cells. Subsequently, the overlap between metabolites significantly affected by an environmental perturbation and metabolites influenced by single-gene deletions (considering only the largest 0.1% of observed metabolic changes) was systematically determined. For each gene deletion, the sum of relative changes among the set of metabolites ($\Omega$) commonly affected by each environmental perturbation ($E$) was calculated according to:

$$S_g = \sum_{\Omega \cap E} Z\text{-score}_i$$

Significance of the overlap was estimated by means of *P*-values obtained using a permutation test:

$$S_g^{\text{perm}} = \sum_{\Omega_{\text{perm}} \cap E} Z\text{-score}_i$$

$$P\text{-value}_g = \frac{(S_g^{\text{perm}} \geq S_g)}{1,000}$$

where $S_{\text{perm}}$ is the permuted score obtained by selecting at random the set of metabolites affected upon gene deletion ($\Omega_{\text{perm}}$). Randomization was performed 1,000 times for *P*-value estimation.

**Growth assay of predicted gene knockouts under deoxycholate stress**

*Escherichia coli* wild-type strain and single-gene deletion mutants with a metabolic phenotype consistent with that of cells under deoxycholate stress (Fig 6A) were cultivated at microtiter scale at

different deoxycholate concentrations (0, 1, 2, 3 mg/ml) in glucose-M9 minimal medium with casein hydrolysate (Fig 6B). Maximum growth rates during exponential growth phase were calculated from triplicate cultivations.

**Growth assay of *Escherichia coli* wild-type strain and enterobactin biosynthesis mutant strains with enterobactin supplementation and deoxycholate stress**

*Escherichia coli* wild-type and mutant strains were cultivated at microtiter scale at different deoxycholate concentrations (0, 1, 2, 3 mg/ml) with varying enterobactin concentration (0, 0.5, 1.5 μM) in glucose-M9 minimal medium (Figs 6C and EV10B). Maximum growth rates during exponential growth phase were calculated from triplicate cultivations.

**Growth assay of *Escherichia coli* wild-type strain under iron limitation and deoxycholate stress**

*Escherichia coli* wild-type strain was cultivated at microtiter scale at different deoxycholate concentrations (0, 1, 2, 3 mg/ml) with varying iron concentration (0.05, 0.15, 0.5, 1, 5, 10, 25, 50 μM) in glucose minimal medium (Fig EV10A). Maximum growth rates during exponential growth phase were calculated from triplicate cultivations.

**Data availability**

The following data are available as separate files online: Profile data for > 34,000 mass spectrometric analysis can be downloaded from http://massive.ucsd.edu/, accession code: MSV000078963. Raw data and modified *z*-scores for positive and negative mode (tab-separated and excel files) can be downloaded from https://www.ebi.ac.uk/biostudies/, accession code: S-BSST5.

**Expanded View** for this article is available online.

### Acknowledgements

This work was in part supported by an ETH postdoctoral fellowship to MZ.

### Author contributions

TF, DCS, and MZ conceived and designed the study, performed the experimental work and computational analysis, and wrote the manuscript. NZ performed computational analysis, supervised the study, and wrote the manuscript. US supervised the study and wrote the manuscript. All authors read and approved the final paper.

### Conflict of interest

The authors declare that they have no conflict of interest.

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
