## [Review Process File · Molecular Systems Biology]

Genomewide landscape of gene–metabolome associations in *Escherichia coli*

Tobias Fuhrer, Mattia Zampieri, Daniel Sévin, Uwe Sauer and Nicola Zamboni

Corresponding author: Uwe Sauer, ETH Zurich

Review timeline:

Submission date:	22 June 2016
Editorial Decision:	25 July 2016
Revision received:	13 December 2016
Accepted:	15 December 2016

Editor: Maria Polychronidou

Transaction Report:

1st Editorial Decision

25 July 2016

Thank you again for submitting your work to Molecular Systems Biology. We have now heard back from the three referees who agreed to evaluate your study. Overall, the reviewers appreciate the value of the presented resource. However, they raise a number of concerns, which should be carefully addressed in a revision of the manuscript. The reviewers' recommendations are clear so there is no need to repeat all the points listed below. In line with comment #5 of reviewer #1, we would like to ask you to make sure that the metabolomics dataset is well documented and easily accessible.

REFEREE REPORTS

Reviewer #1:

The present manuscript reports what is to my knowledge the 1st genome-wide metabolomics analysis of single gene deletions. The work is conducted in *E. coli*, a common model organism with a vibrant community of metabolism researchers. The direct infusion metabolomics approach, while subject to many types of interferences, provides the required throughput and is a good match to the genome-scale of the task. Using the resulting data, the authors provide evidence for local effects of gene knockouts on metabolism, not only for enzymes but also for transcription factors (via their enzyme targets) and non-metabolic genes that interact with metabolic genes. They also show strong clustering of related proteins based on their metabolome profiles. And they annotate YhbC is a dihydrotate hydrolase. Overall, the community has been waiting some time for a genome-wide knockout library to be analyzed by metabolomics, and I think there will be great interest in and widespread use of the data.

Major concerns:

1. The metabolomics approach is subject to mis-identification and mis-quantitation, especially due to in-source fragmentation. While this does not detract from the suitability of use of this approach for the present study, the authors need to clearly acknowledge these limitations/risks.

2. Many of the figures conveyed essentially nothing to me, either because they were unreadably small or they had no message. In this problem category, I would put 1B, 1C, 1D, 2B, 4A, 4B. The authors need to come up with a compelling set of figures that each convey a clear scientific message (or specific data), as opposed to just a pretty picture.

3. The follow-up work on YhbC is unacceptable quality. The only tangible new knowledge in the paper (separate from the copious and important new data resource) is that YhbC is a dihydroxotrate hydrolase. Accordingly, this needs to be nailed. The authors minimally need to meet basic biochemistry standards: reporting of specific activities in standard units not random amounts of protein on a random MS scale, showing both depletion of substrate and accumulation of product and whether those are stoichiometric, determining K_m , etc. On the intellectual front, this enzyme is quite puzzling, since the cyclization of carbamoyl-aspartate to DHO is thought to be spontaneous (as most intramolecular peptide bond cyclizations are). If this enzyme really catalyzes the reversible reaction, what is the energy source? It is thermodynamically impossible for PyrC and YhbC to catalyze the identical reaction in opposite directions under the same conditions. Can the authors learn anything from when this enzyme is expressed or when it is essential in terms of its physiological role?

4. The final story about deoxycholate was completely lost on me. From the sentence about "mimics gastric stress caused by bile acids" I was confused and I stayed confused through the end of the paper. I cannot say more about the science, because I really do not know what I was supposed to take away from this.

5. I think that the most valuable aspect of this paper should be the resource of metabolomics data for all of the single-gene deletions. In going through the supplement, I did not find this table readily. The quality of this table is paramount to the long-term impact of the paper-- it needs to be easy to find and very well annotated.

Minor issues:

1. I do not love the title, and do not think that the paper is really about "gene-metabolite associations" or "landscapes." It is about metabolomics analysis of a full genome knockout collection.

2. I certainly object to the claims of the paper is unprecedented. What is nice about this paper is that actually it is very well preceded by all sorts of other genome-scale work, especially phenotyping of genome-scale knockout collections, activities that in the past have proven extremely valuable and to make great resources.

3. In Fig 1E, the authors restrict the analysis to genes with at least one significant metabolic change. What fraction of genes showed such a change? If most, then the analysis should be fine, but if it is a minority, the authors probably need to think of a way to avoid such selection bias.

4. While I understand the locality analysis after reading the supplement, I feel that it could have been much better explained in the paper. Looking at figure 2A, I had the idea that all 5 degrees were significant (they aren't radically different from one another), whereas I see in the supplement that the degree 3, 4, 5 are really the control for degree 1, and there is no control to show that degree 5 separation is different or the same as infinite separation.

5. It would be helpful for the authors to provide some examples of non-metabolic enzyme - metabolite associations. What are these typically?

Reviewer #2:

The manuscript by Fuhrer et al. addresses a topical question in systems biology - gene-metabolite (concentration) relationships. To address this, the authors use an unbiased mass-spectrometry approach to characterize metabolomes of the mutant strains in the E. coli single gene knockout library. The results reveal a comprehensive network of gene-metabolite associations that not only recaptures the known and expected links but also presents novel hypotheses, few of which are followed-up in detail. Overall, the study provides a valuable resource and a conceptual step ahead in understanding genotype-(metabolic) phenotype relations. There are, however, a few questions/concerns that need to be addressed.

1. Network proximity analysis: The authors exclude highly connected metabolites from their distance calculations. This approach is not biologically warranted for an unbiased systems study like this (which aims at revealing connections that are mediated through yet unknown mechanisms). Cofactors like NADPH or ATP, for example, can wield a widespread influence on metabolic network connecting distant pathways like PP pathways and TCA cycle. This could potentially explain the discussed cases like malate-aro/pur connection. I therefore recommend performing an additional analysis including physiologically important co-factors in distance calculations. (Use of flux coupling network could also make the analysis more comprehensive).

2. Metabolite annotations: Though authors acknowledge the possible uncertainty in metabolite annotations, the discussion lacks this point. Furthermore, it is unclear whether the pre-processing filters out highly correlated ions that could result from in-source fragmentation or other artifacts.

3. Biological replicates: Perhaps I missed this but what is the R (or rho) value of the responses between the two biological replicates?

4. Semantics:

i) Abstract: "enzyme deletions"

ii) Introduction: "...traits by which genes can influence..." (Traits are not the means by which the genes influence something)

Reviewer #3:

A major challenge, and perhaps the greatest challenge, in the field of microbiology is to better understand the role of the numerous genes of unknown function, even in extensively studied organisms like E. coli. Screening gene knockouts has yielded a great deal of knowledge in this space, but is typically limited by the screening of all gene knockouts against a limited set of factors (e.g., the presence/absence of a specific carbon source). In "Genome-wide landscape of gene-metabolome associations in Escherichia coli" the authors perform metabolomics on thousands of metabolites across the E. coli gene knockout collection to associate genes with metabolic alterations. The result provides a rich collection of information to relate genes to each other and to compare the metabolic changes to environmental stimuli and identify similar metabolic changes associated with gene knockouts. I think the manuscript is an excellent example of how to address this difficult problem of microbial gene annotation. I have only a few minor comments.

1) The utility of this method is somewhat limited by the difficulty of comparing metabolomics data between platforms and laboratories. While in theory a new laboratory would test an additional environmental perturbation and compare their metabolomics result with the dataset from this manuscript, in practice it is quite difficult to do in the context of mass spec, which limits the utility of the generated metabolomics dataset. However, this is simply a limitation of the technology in its current state.

2) It seems the authors use a different culture media in the initial knock-out screen and the later environmental perturbation experiments. I could not find a justification for this switch but it would potentially complicate metabolomics results greatly to have two different background media (e.g., I believe one has casein and the other does not). Although the authors compare the environmental

samples to a control sample with no perturbation on the same media, it is still a little worrisome to switch media like this.

3) prediction of orphan gene

In the main text, this section lacks sufficient methodological information to interpret where they numbers and results come from. While I agree that a lot of the details can be placed in the supplement, I think the balance is a little off here and at least a slightly expanded and intuitive explanation of the methods needs to be brought into the main text.

Typo line 633: annotated should be annotated

1st Revision - authors' response

13 December 2016

Reviewer #1:

The present manuscript reports what is to my knowledge the 1st genome-wide metabolomics analysis of single gene deletions. The work is conducted in *E. coli*, a common model organism with a vibrant community of metabolism researchers. The direct infusion metabolomics approach, while subject to many types of interferences, provides the required throughput and is a good match to the genome-scale of the task. Using the resulting data, the authors provide evidence for local effects of gene knockouts on metabolism, not only for enzymes but also for transcription factors (via their enzyme targets) and non-metabolic genes that interact with metabolic genes. They also show strong clustering of related proteins based on their metabolome profiles. And they annotate YhbC is a dihydroxatate hydrolase. Overall, the community has been waiting some time for a genome-wide knockout library to be analyzed by metabolomics, and I think there will be great interest in and widespread use of the data.

We would like to thank the referee for the positive review and the appreciation of the value of the data for the research community! Please find the point-by-point response below.

Major concerns:

1. The metabolomics approach is subject to mis-identification and mis-quantitation, especially due to in-source fragmentation. While this does not detract from the suitability of use of this approach for the present study, the authors need to clearly acknowledge these limitations/risks.

In principle we agree with the reviewer that there's a latent risk of mis-quantification and mis-annotation, but not because of spontaneous fragmentation. We have extensive experience and measurements to characterize the extent to fragmentation on pure standards on exactly the same instrument. We observed both in source and post-source fragmentation, i.e. through cooling of ions in the high-pressure collision cell before ions beam focusing and TOF. The hardware is operated and tuned to minimize such problems to only less than a few percent. Hence, degradation fragments are observable only for abundant metabolites. At the level of software, we actually check for common derivatives (loss of P, PPi, CO₂, NH₃, H₂O) throughout the data and use the correlation score between intensities of the molecular ion and that of the putative fragment to check the likelihood that it's indeed a fragment and not a different molecular ion. All of this is convoluted in the annotation score that we report (referenced in the main text as Fuhrer et al 2011). Regardless of this analysis, for any given measured ion we enumerate all possible matches.

The main issue is mis-quantification and miss-annotation due to peak crowding or presence of unknown unknowns. We added an explicit comment at the very beginning of the results section.

2. Many of the figures conveyed essentially nothing to me, either because they were unreadably small or they had no message. In this problem category, I would put 1B, 1C, 1D, 2B, 4A, 4B. The authors need to come up with a compelling set of figures that each convey a clear scientific message (or specific data), as opposed to just a pretty picture.

We thank the reviewer for pointing this out. Complex figures were split. We improved the quality of the pictures in terms of size and readability. We also worked over all legends to clarify and explain better what is shown.

3. The follow-up work on YhbC is unacceptable quality. The only tangible new knowledge in the paper (separate from the copious and important new data resource) is that YhbC is a dihydroxate hydrolase. Accordingly, this needs to be nailed. The authors minimally need to meet basic biochemistry standards: reporting of specific activities in standard units not random amounts of protein on a random MS scale, showing both depletion of substrate and accumulation of product and whether those are stoichiometric, determining K_m , etc. On the intellectual front, this enzyme is quite puzzling, since the cyclization of carbamoyl-aspartate to DHO is thought to be spontaneous (as most intramolecular peptide bond cyclizations are). If this enzyme really catalyzes the reversible reaction, what is the energy source? It is thermodynamically impossible for PyrC and YhbC to catalyze the identical reaction in opposite directions under the same conditions. Can the authors learn anything from when this enzyme is expressed or when it is essential in terms of its physiological role?

We appreciate the comment that highlighted a superficial discussion of the results. The apparent confusion originates from the fact that the *in vitro* the reaction can be driven in either direction (without any cofactor) by excess substrate. *In vivo*, the pathway is likely to work in the canonical direction.

Overall, we regret that we can't provide the information to the biochemical detail outlines by the Reviewer (K_m etc). One single, very detailed example is of little relevance and doesn't change much in the economy of a genome-wide study. As highlighted in the text and by the reviewers, the main contribution is the resource and the analysis that points to massive presence of "distal" gene-metabolite interactions. Unfortunately, the molecular origin of such links remains largely unexplained at the moment, but the connections are statistically significant.

The function predictions and stress response are examples on how the association map can be queried to generate hypothesis on gene function. Because of the nature of the data, the complexity of the response, and the incomplete metabolome coverage, the prediction related to the pathway or functional level. This aspect is reflected in the detailed discussion of the 70+ prediction that we inferred using the CLR algorithm. To avoid a misrepresentation of the results and merits of the prediction, we decided to remove the incomplete analysis of the YhbC case.

4. The final story about deoxycholate was completely lost on me. From the sentence about "mimics gastric stress caused by bile acids" I was confused and I stayed confused through the end of the paper. I cannot say more about the science, because I really do not know what I was supposed to take away from this.

We thank the reviewer for pointing out that this part was unclear. We have now clarified the paragraph about the metabolic response to environmental perturbations and in particular to deoxycholate stress.

5. I think that the most valuable aspect of this paper should be the resource of metabolomics data for all of the single-gene deletions. In going through the supplement, I did not find this table readily. The quality of this table is paramount to the long-term impact of the paper-- it needs to be easy to find and very well annotated.

We thank the reviewer and editor for pointing this out and we totally agree that accessibility of the metabolomics data is crucial. The information of how and where to

access the raw data online is now contained in the main text, at the end of Materials and Methods in the new section "Data availability".

Minor issues:

1. I do not love the title, and do not think that the paper is really about "gene-metabolite associations" or "landscapes." It is about metabolomics analysis of a full genome knockout collection.

Apparently the reviewer prefers a title that describes what was done, whereas we prefer a title that describes what was achieved. The main achievement in our view is the extensive map of gene-metabolite associations that will be made available to the community. The summary of these interactions is actually a map of a landscape that we never looked at in this unbiased fashion. Hence, we prefer to stick to our title. We do describe very clearly what was done in the abstract.

2. I certainly object to the claims of the paper is unprecedented. What is nice about this paper is that actually it is very well preceded by all sorts of other genome-scale work, especially phenotyping of genome-scale knockout collections, activities that in the past have proven extremely valuable and to make great resources.

We never intended to claim that we performed the first genome-scale analysis of any type, but we are not aware of another metabolomics study at this scale. Unprecedented referred to the fact that phenotypic studies generally have one or very few read outs (such as growth rate), while we provide multiple readouts for each case. In some ways the reviewer even supports our claim by saying that the field has been waiting for this. Nevertheless, we removed the term entirely and revised the text for clarification.

The key advantage of our metabolomics approach over previous phenotypic studies is the capacity to detect responses even in the absence of growth phenotypes, thereby increasing functional readouts by orders of magnitude compared to classical phenotypic screens. In this sense, our compendium of >7000 relative metabolite ion levels in >3800 mutants is unprecedented.

3. In Fig 1E, the authors restrict the analysis to genes with at least one significant metabolic change. What fraction of genes showed such a change? If most, then the analysis should be fine, but if it is a minority, the authors probably need to think of a way to avoid such selection bias.

Good point. If we apply a significance cutoff of $|\text{absolute}(z\text{-score})| > 5$, we find that about 50% of the gene knock-outs exhibited at least one metabolic change. Hence, the analysis was based on considering half of the genes and is therefore representative.

4. While I understand the locality analysis after reading the supplement, I feel that it could have been much better explained in the paper. Looking at figure 2A, I had the idea that all 5 degrees were significant (they aren't radically different from one another), whereas I see in the supplement that the degree 3, 4, 5 are really the control for degree 1, and there is no control to show that degree 5 separation is different or the same as infinite separation.

Notably, the larger the distance between enzyme and metabolite, the fewer pairs can be retrieved. We here show the statistics up to a distance of five because we need a sufficient number of enzyme-metabolite pairs to draw a reliable estimate of p-value distribution. Moreover, the distribution of p-values already shows clearly that after a distance of 2 there is no enrichment of larger metabolic changes in the proximity of the deleted enzyme, and distribution of p values become uniform.

5. It would be helpful for the authors to provide some examples of non-metabolic enzyme - metabolite associations. What are these typically?

We apologize for the confusion. What we meant are non-enzymatic gene-metabolite associations based on non-metabolic genes listed in the ECID database (see Figure EV6C).

Examples of typical non-metabolic associations include maturation factors (e.g. YgfY, recently annotated as SdhE) affecting the reactant levels of the respective enzyme function, transporters (e.g. brnQ, branched chain amino acid transporter) affecting intermediates of the respective degradation pathways or transcriptional regulators (e.g. puuR, DNA binding transcriptional repressor sensing putrescine). The examples were added to Figure 1 in the case of BrnQ and PuuR while YgfY is reported in Figure EV8.

Reviewer #2:

The manuscript by Fuhrer et al. addresses a topical question in systems biology - gene-metabolite (concentration) relationships. To address this, the authors use an unbiased mass-spectrometry approach to characterize metabolomes of the mutant strains in the E. coli single gene knockout library. The results reveal a comprehensive network of gene-metabolite associations that not only recaptures the known and expected links but also presents novel hypotheses, few of which are followed-up in detail. Overall, the study provides a valuable resource and a conceptual step ahead in understanding genotype-(metabolic) phenotype relations. There are, however, a few questions/concerns that need to be addressed.

Firstly, we would like to thank the referee for the positive feedbacks and comments. Please find the point-by-point response below.

1. Network proximity analysis: The authors exclude highly connected metabolites from their distance calculations. This approach is not biologically warranted for an unbiased systems study like this (which aims at revealing connections that are mediated through yet unknown mechanisms). Cofactors like NADPH or ATP, for example, can wield a widespread influence on metabolic network connecting distant pathways like PP pathways and TCA cycle. This could potentially explain the discussed cases like malate-aro/pur connection. I therefore recommend performing an additional analysis including physiologically important co-factors in distance calculations. (Use of flux coupling network could also make the analysis more comprehensive).

Exclusion of cofactors or ‘currency metabolites’ is quite common in the field and was shown to be necessary to obtain meaningful predictions (Hancock et al, 2012; Kim et al, 2015; Noor et al, 2010; Ravasz et al, 2002; Schulz et al, 2014). This is compatible with the view that removing a reaction that uses ATP, NADPH, or the C1 pool is unlikely to affect other reactions that use the same cofactor since it’s availability isn’t compromised. To our knowledge, cross-pathway interactions mediated by cofactors are mostly allosteric and not catalytic (e.g. NADPH-repression of the Zwf, ATP binding to Pfk).

This is different if the deletion leads to a reduction of the “active” form of the cofactor. This happens if the main cellular processes responsible for cofactor production or regeneration are disturbed. In our data, we observed that gene deletions in coenzyme transport and metabolism, nucleotide transport and metabolism, signal transduction mechanisms and transcription tend to induce numerous metabolic changes (new Figure EV7). This indicates that all these processes are important and mutations lead to pleiotropic effects. This isn’t surprising, because such mutations prevent the generation of the cofactors, which likely becomes limiting for all coupled reactions. The metabolic consequences of cofactor depletion are widespread. They don’t explain the emergence of rather specific association between KOs and apparently distal metabolites.

We now included a new figure EV7, and discussed this aspect in the main text.

References:

Hancock T, Wicker N, Takigawa I, Mamitsuka H (2012) Identifying neighborhoods of coordinated gene expression and metabolite profiles. PLoS one 7: e31345

Kim T, Dreher K, Nilo-Poyanco R, Lee I, Fiehn O, Lange BM, Nikolau BJ, Sumner L, Welti R, Wurtele ES, Rhee SY (2015) Patterns of metabolite changes identified from large-scale gene perturbations in Arabidopsis using a genome-scale metabolic network. *Plant physiology* 167: 1685-1698

Noor E, Eden E, Milo R, Alon U (2010) Central carbon metabolism as a minimal biochemical walk between precursors for biomass and energy. *Mol Cell* 39: 809-820

Ravasz E, Somera AL, Mongru DA, Oltvai ZN, Barabasi AL (2002) Hierarchical organization of modularity in metabolic networks. *Science* 297: 1551-1555

Schulz JC, Zampieri M, Wanka S, von Mering C, Sauer U (2014) Large-scale functional analysis of the roles of phosphorylation in yeast metabolic pathways. *Sci Signal* 7: rs6

2. Metabolite annotations: Though authors acknowledge the possible uncertainty in metabolite annotations, the discussion lacks this point. Furthermore, it is unclear whether the pre-processing filters out highly correlated ions that could result from in-source fragmentation or other artifacts.

We thank the reviewer for pointing this out. We added a sentence discussing the impact of potential miss-annotations. In the published method paper (referenced in the main text as Fuhrer et al 2011) we investigated generally observed highly correlating ions and identified the respective frequent mass shifts mostly as neutral losses such as for example H₂O or NH₃ as well as adducts such as +H₂PO₄Na or +H₂PO₄K. These frequently occurring mass shifts are included in the annotation procedure.

In addition the voltage settings of the mass-spectrometry method were chosen to prevent in-source fragmentation and other artifacts as much as possible.

3. Biological replicates: Perhaps I missed this but what is the R (or rho) value of the responses between the two biological replicates?

The mean rho is 0.1059. This information wasn't included because rho is not a good benchmark for reproducibility given that we measured thousands of features but only a little fraction is significantly associated to a given genotype. The effect of metabolic "markers" is eclipsed by the noise associated to the overwhelming number of non-markers and z-score normalization. To bypass this issue, we assessed reproducibility by estimating a background distribution of Z-scores using the biological replicates. The analysis and results are now reported in figure EV2A.

4. Semantics:

- i) Abstract: "enzyme deletions"

Corrected.

- ii) Introduction: "...traits by which genes can influence..." (Traits are not the means by which the genes influence something)

Corrected.

Reviewer #3:

A major challenge, and perhaps the greatest challenge, in the field of microbiology is to better understand the role of the numerous genes of unknown function, even in extensively studied organisms like E. coli. Screening gene knockouts has yielded a great deal of knowledge in this space, but is typically limited by the screening of all gene knockouts against a limited set of factors

(e.g., the presence/absence of a specific carbon source). In "Genome-wide landscape of gene-metabolome associations in *Escherichia coli*" the authors perform metabolomics on thousands of metabolites across the *E. coli* gene knockout collection to associate genes with metabolic alterations. The result provides a rich collection of information to relate genes to each other and to compare the metabolic changes to environmental stimuli and identify similar metabolic changes associated with gene knockouts. I think the manuscript is an excellent example of how to address this difficult problem of microbial gene annotation. I have only a few minor comments.

We thank the reviewer for the very positive review. Please see below for the point-by-point responses.

1. The utility of this method is somewhat limited by the difficulty of comparing metabolomics data between platforms and laboratories. While in theory a new laboratory would test an additional environmental perturbation and compare their metabolomics result with the dataset from this manuscript, in practice it is quite difficult to do in the context of mass spec, which limits the utility of the generated metabolomics dataset. However, this is simply a limitation of the technology in its current state.

The preconditions aren't overly strict, because (in our experience) it's sufficient to compare qualitative changes for features that can be matched with deprotonated/protonated metabolites. We already have done (successfully) this with data generated on different MS instruments that offer a similar or better coverage of metabolism and with cells grown in different media.

2. It seems the authors use a different culture media in the initial knock-out screen and the later environmental perturbation experiments. I could not find a justification for this switch but it would potentially complicate metabolomics results greatly to have two different background media (e.g., I believe one has casein and the other does not). Although the authors compare the environmental samples to a control sample with no perturbation on the same media, it is still a little worrisome to switch media like this.

We thank the reviewer for raising the point on potential drawbacks from comparing different. In the screening, we opted for a rich medium with casein amino acids to (i) better reflect the variety of natural substrates, (ii) compensate auxotrophies of mutants lacking genes that are essential for growth on glucose, (iii) alleviate growth rate differences. The exact composition of the screening medium isn't known. For instance, it also contains nucleobases (xanthine and hypoxanthine) and other peaks we can't assign.

The follow ups were done with glucose minimal medium to reduce potential confounding variables. This is common practice, but could be a source of difference in results. For the cases we discussed, this change doesn't seem to be relevant. It is also plausible that in our examples of environmental perturbations such as phosphate or sulphur limitation, the presence of amino acids would not bias our predictions. These results actually speak in favor of the resource we generated. We expect that for certain questions involving amino acids metabolism one would have to use a rich medium to compare responses. We now clarify this limitation in the text.

3. prediction of orphan gene: In the main text, this section lacks sufficient methodological information to interpret where they numbers and results come from. While I agree that a lot of the details can be placed in the supplement, I think the balance is a little off here and at least a slightly expanded and intuitive explanation of the methods needs to be brought into the main text.

Also requested by the Editor, all Materials and Methods as well as Results from the Supplements have been moved to the main text and merged wherever necessary.

4. Typo line 633: annotated should be annotated

Corrected

2nd Editorial Decision

15 December 2016

Thank you for sending us your revised manuscript. We have now heard back from reviewer #2 who was asked to evaluate the study. The reviewer is satisfied with the modifications made and I am pleased to inform you that your paper has been accepted for publication.

Reviewer #2:

The authors have addressed my comments in the revised version.

Corresponding Author Name: Uwe Sauer

Journal Submitted to: MSB

Manuscript Number: MSB-16-7150R